# Poison Center Surveillance of Occupational Incidents with Hazardous Materials (2016–2023): Insights for Risk Mitigation and Incident Preparedness

**DOI:** 10.3390/ijerph22020158

**Published:** 2025-01-25

**Authors:** Anja P. G. Wijnands, Arjen Koppen, Irma de Vries, Dylan W. de Lange, Saskia J. Rietjens

**Affiliations:** 1Dutch Poisons Information Center, University Medical Center Utrecht, Utrecht University, P.O. Box 85500, 3508 GA Utrecht, The Netherlands; a.koppen@umcutrecht.nl (A.K.); vriesdi7@gmail.com (I.d.V.);; 2Department of Intensive Care Medicine, University Medical Center Utrecht, Utrecht University, P.O. Box 85500, 3508 GA Utrecht, The Netherlands

**Keywords:** HAZMAT incidents, chemical incidents, surveillance, public health risks, poison information centers, prevention, risk mitigation strategies

## Abstract

Incidents involving hazardous materials (HAZMAT incidents) can impact human health and the environment. For the development of risk mitigation strategies, it is essential to understand the circumstances of such incidents. A retrospective study (2016–2023) of acute occupational HAZMAT incidents involving multiple patients (>1, including workers, emergency responders and bystanders) reported to the Dutch Poisons Information Center was conducted. We only included incidents that occurred during the performance of work or as a result of a disruption of a work-related process. Patient characteristics, exposure circumstances (such as the substances involved, chemical phase, and type of release (e.g., spill/release or fire/explosion)) and business classes were analyzed to identify risk factors. From 2016 to 2023, the DPIC was consulted about 516 HAZMAT incidents. Inhalation was the most common route of exposure (89%). Patients were often exposed to chemical asphyxiants *(n* = 156) and acids (*n* = 151). Most incidents occurred in fixed facilities (*n* = 447), while 49 incidents occurred during transport. The primary cause was a spill/release (*n* = 414), followed by a fire/explosion (*n* = 65). Most patients were exposed to a gas/vapor (*n* = 421), followed by a liquid (*n* = 59) or solid (*n* = 28). Incidents frequently occurred in industry (20%). The majority of patients reported mild to moderate health effects. Surveillance data on HAZMAT incidents are essential for incident preparedness. Poison Center data can help identify risk factors, which can be used to develop risk mitigation strategies to prevent future incidents.

## 1. Introduction

Incidents involving the release of hazardous materials (HAZMAT incidents), like chemical spills, leaks, fires, and the intentional release of toxic substances, may have significant adverse effects on human health, the environment and society, causing casualties, economic losses, and ecological damage [1,2,3,4,5,6,7,8,9,10]. The Centre for Research on the Epidemiology of Disasters reported that between 2000 and 2021, technological disasters (i.e., industrial accidents (such as chemical spills, explosions, and gas leaks) and transport accidents) affected 2,638,985 people in total and caused USD 63,178 million worth of losses worldwide [11]. According to the World Health Organization (WHO), an estimated 65,000 people died due to technological events between 2009 and 2018 [12]. 

HAZMAT incidents often have an occupational origin and can occur during several activities, such as production, storage, transportation, use, and waste disposal [2]. In the event of such incidents, not only workers but also first responders and the general public could be at risk of harm [13,14,15]. 

In the Netherlands, numerous organizations play an important role in the management of HAZMAT incidents. Such incidents are frequently handled through the collaborative efforts of the emergency services (such as the police, fire brigade, and ambulance service), local and national governmental bodies, environmental organizations, and the industry itself. The Dutch Poisons Information Center (DPIC) fulfills several important roles in the response to HAZMAT incidents in the Netherlands. An important task of the DPIC is to provide a 24/7 telephone service offering expert advice to healthcare professionals regarding the diagnosis and treatment of exposed patients. We recently showed that the annual number of calls related to acute occupational exposure to dangerous substances in workers almost doubled from 475 in 2016 to 936 in 2022. Most of these occupational exposures involved small-scale incidents, where only one worker was exposed (i.e., 5128 incidents from 2016 to 2022). However, there were also numerous incidents in which more than one person was exposed (around 40–70 incidents per year from 2016 to 2022) [16]. In addition to providing expert advice on health risks and medical treatment on small-scale occupational exposures, the expertise of the DPIC is also regularly deployed in larger incidents involving hazardous substances. These incidents frequently involve the exposure of multiple individuals simultaneously, and/or the potential for the dispersion of hazardous substances. 

Surveillance data offer crucial insights into the prevalence and circumstances of HAZMAT incidents, which are essential for effective health and safety management [17,18,19,20,21,22]. In the Netherlands, multiple data sources document work-related fatalities and injuries [23,24,25,26,27]. For instance, VeiligheidNL reported that in 2017, an estimated 48,200 patients visited an emergency department following a workplace injury. Among these incidents, only a minority involved chemical exposure (e.g., the skin or eyes; approximately 600 ED visits) or intoxication (approximately 500 ED visits). However, the causes of these exposures were not listed [25]. Additionally, the Dutch Institute for Public Health and the Environment (RIVM) analyzes the nature, scale, and causes of incidents at companies handling large quantities of dangerous substances (Seveso facilities). Between 2019 and 2021, fourteen incidents involved the release of hazardous substances. One person likely sustained permanent injuries from a chemical burn, while other victims (twenty-one in six separate incidents) suffered temporary injuries such as breathing problems, skin irritation, and burns [24].

Despite the presence of several authorities in the Netherlands that report on occupational incidents, a comprehensive surveillance system that reports information on all types of occupational incidents involving hazardous materials (HAZMAT incidents) is currently lacking. The DPIC receives numerous calls about occupational incidents, including those with relatively minor health consequences that are not included in regular Dutch injury statistics and are therefore considered supplementary [16]. Although these incidents may seem minor, they can be precursors to more significant incidents and provide valuable learning opportunities. Furthermore, these data can also be beneficial for surveillance purposes.

This report summarizes acute hazardous chemical incidents involving more than one patient reported to the DPIC between 2016 and 2023. All incidents involving occupational exposure (e.g., workers were exposed) and incidents in which the exposure was caused by a work process (e.g., workers and/or bystanders were exposed) are studied. The aim of this study is to examine the characteristics (number and age of patients, health effects, treatment recommendations) and exposure circumstances (routes of exposure, substances involved, causes, business classes involved) of these incidents. A better understanding of the exposure circumstances of these chemical incidents can improve prevention strategies and enhance preparedness for future incidents.

## 2. Materials and Methods

### 2.1. Study Design and Study Population 

A retrospective analysis was conducted on cases involving multiple patients reported to the Dutch Poison Information Center (DPIC) between 1 January 2016 and 31 December 2023. This study only included cases in which the exposure to the hazardous substance(s) occurred during the performance of professional work (occupational exposures) or as a result of a disruption of a work-related process (such as a work-related fire, explosion, or leakage). Workers, emergency responders, and bystanders may all be exposed during these type of HAZMAT incidents. The study was limited to acute exposure, which was defined as a single or short-term exposure with a maximum duration of exposure of one day (a research flow chart is presented in Figure 1). 

### 2.2. Data Sources and Collection

The DPIC offers a 24/7 telephone service to provide expert advice to healthcare professionals in the Netherlands on the diagnosis and treatment of patients exposed to potentially toxic substances. The data have been extracted from the DPIC database, in which anonymous information is recorded in a standardized data format to ensure consistent collection of data. Multiple inquiries regarding the same incident were treated as a single inquiry. The accredited Medical Research Ethics Committee of the University Medical Center Utrecht did not consider the Dutch Medical Research Involving Human Subjects Act to be applicable to this study.

### 2.3. Variables and Classifications

For each incident, the following variables were examined: type of enquirer (such as a general practitioner, emergency department, ambulance, Municipal Health Service, member of the public), patient characteristics (number of patients, age), exposure characteristics (route of exposure, involved substance(s), causes of the incident, business classes involved), symptoms reported at the time of DPIC consultation, and treatment recommendations (wait-and-see policy, examination by a physician, or hospital observation). The involved substances were identified using the product information provided by the caller. In incidents where specific product names were known, the legally supplied information on the chemical composition of hazardous products provided by the company to the DPIC was used (classification, labeling and packaging regulation [28]). The incidents were classified based on four different factors. Firstly, all incidents were categorized as either accidental or intentional. Secondly, all accidental incidents were classified according to whether they occurred during transport (by road, water (including harbors) or air), within a fixed facility, or in other contexts. Thirdly, all accidental incidents were classified according to the cause of the release, which could be the result of a fire or explosion, a spill or leakage, or some other cause. Finally, the accidental incidents were stratified according to the chemical phase of the substance(s) involved in the incident, distinguishing between exposure to gases or vapors, liquids and solids. The classification system was, in part, based on information from the Hazardous Substances Emergency Events Surveillance (HSEES) of the US Agency for Toxic Substances and Disease Registry (ATSDR). The HSEES system was used from January 1991 to September 2009 to describe the public health consequences of chemical incidents in the US [29,30,31]. Business classes were categorized using the Standard Business Classification List of Statistics Netherlands (CBS) published by Kruiskamp (2008) [32].

### 2.4. Analysis

We used descriptive statistics, such as percentages, medians, interquartile ranges, and full ranges, to summarize the characteristics of our dataset. Calculations and data analysis were conducted using Microsoft Excel® for Microsoft 365 MSO (version 2402 Build 17328.20612), R Studio® (version 2024.09.1 Build 394 for Windows), and IBM SPSS Statistics (version 29.0.1). 

## 3. Results

### 3.1. Number of Acute Occupational HAZMAT Incidents Reported to the Dutch Poisons Information Center (2016–2023)

In total, 516 acute occupational HAZMAT incidents were reported to the DPIC over the 8-year study period. The number of HAZMAT incidents fluctuated, with a range of 52 to 81 incidents per year over the course of the study period (Figure 2). 

### 3.2. Number of Patients Involved and Patient Characteristics

The exact number of patients involved in the 516 HAZMAT incidents reported to the DPIC was not known. In many cases, we were called by a healthcare professional who was treating one specific patient. The other patients involved in the incident did not always seek medical assistance, or in some cases they were treated by another healthcare professional who did not contact the DPIC. Only cases in which the caller mentioned that more than one individual had been exposed were included in our study. In total, at least 1840 patients were involved, but the number is most probably higher.

The number of patients per incident ranged from 2 (213 incidents) to 50 (1 incident). Most incidents (*n* = 371, 72%) were relatively small, involving up to five patients. There were 21 (4%) incidents involving between 6 and 10 patients, 15 (3%) involving between 11 and 20 patients, and 6 (1%) involving more than 20 patients. In 103 incidents (20%), the exact number of patients involved was unknown.

For 517 patients, the precise age was known. For these patients, the median age was 28 years (interquartile range: 25 years, full range: 4–90 years). One hundred and sixty-one children younger than 18 years of age were involved in the 516 incidents studied. The age of 139 children was known. For these children, the median age was 10 years (interquartile range: 4 years, full range: 4–17 years).

### 3.3. Type of Enquirer and Treatment Recommendations

The most common initial contact with the DPIC was made by general practitioners (41%, 211 calls), followed by emergency departments (17%, 89 calls), ambulance services (11%, 57 calls), Municipal Health Services (7%, 35 calls), and members of the public (4%, 20 calls). In a large number of incidents, we received multiple successive calls. Follow-up calls were mainly made by general practitioners (44 calls), Municipal Health Services (23 calls), and hospital physicians (21 calls). 

A treatment recommendation was given for 1441 patients. A wait-and-see approach was recommended for 736 patients, with instructions to contact a physician if symptoms worsened or further symptoms occurred. A total of 559 patients were advised to see a doctor for further examination. Hospital observation and/or treatment were recommended for 146 patients. Calls from emergency departments and hospital physicians indicate that 465 patients visited a hospital for examination, observation, or treatment.

### 3.4. Exposure Characteristics (Route of Exposure and Substances Involved)

Patients were frequently exposed via multiple routes. The most common route of exposure was inhalation (89%), followed by ocular (19%), dermal (16%), and oral exposure (5%). In the 26 incidents involving oral exposure, 11 incidents also involved inhalation and/or eye and/or dermal contact. In these cases, oral exposure was primarily the result of splashes to the face or spraying. However, there were also 15 incidents involving ingestion as the only route of exposure, predominantly following the consumption of contaminated water or beverages.

The substances involved in HAZMAT incidents are diverse, and patients may be exposed to multiple substances simultaneously. An overview of the most common substances involved is provided in Table 1.

### 3.5. Causes of HAZMAT Incidents Reported to the Dutch Poisons Information Center (2016–2023)

In order to provide a more detailed description of the scenarios and to identify potential risk factors, the 516 incidents reported to the DPIC between 2016 and 2023 were categorized (Figure 3). 

The majority of incidents were accidental (*n* = 508, 98%). Eight incidents (2%) were classified as intentional, including exposure to suspicious powder letters (*n* = 2), pepper spray (*n* = 2), or fire extinguishers (*n* = 1), or the deliberate addition of chemicals to food or beverages (*n* = 3).

Most accidental incidents occurred at a fixed facility (*n* = 447, 88%). Forty-nine incidents (10%) happened during transport. Transport incidents can be divided into incidents that occur during transport by water (*n* = 22, including incidents in harbors), road (*n* = 11), air (*n* = 4), rail (*n* = 3), or container storage (*n* = 9). Accidental exposure was predominantly the result of a spill or release (*n* = 414, 82%), followed by a fire or explosion (*n* = 65, 13%). Most patients were accidentally exposed to a gas/vapor (*n* = 421, 83%) with fewer exposures to a liquid (*n* = 59, 12%) or solid (*n* = 28, 6%).

### 3.6. Business Classes Involved in HAZMAT Incidents Reported to the Dutch Poisons Information Center (2016–2023)

Most incidents occurred in industry (20%), the transport sector (12%), health and welfare care (10%), and the building and installation industry (9%) (Table 2). However, a significant number of incidents also occurred in other business classes. 

All 104 industrial incidents happened in a fixed facility, and the majority of the incidents (96 incidents, 92%) were caused by a spill or release of hazardous substances (release of gas/vapor (*n* = 71), liquid (*n* = 18), or solid (*n* = 7)). The hazardous substances involved were very diverse, with acids (*n* = 15) and alkalis (*n* = 10) being the most common. The following examples of causes were mentioned: overturning of storage containers containing hazardous substances, defective ventilation, rupturing of a pipeline, not working according to protocols, or not using adequate personal protective equipment (PPE). In the transport and storage sector, the majority of accidents were also the result of a spill or release, accounting for 56 incidents (90% of cases). Many accidents occur when loading or unloading cargo or containers, or when working near damaged containers. Accidents during the transportation (water, rail, road, air) of hazardous substances were also reported. In 14 incidents, patients were exposed to pesticides. Of these, 11 incidents involved exposure to phosphine gas after containers were opened or during transport, primarily during maritime transport. In the health and welfare sector, the circumstances were very diverse, including incidents such as dropping bottles (*n* = 9), fires (*n* = 8), defects in equipment (*n* = 5), cleaning accidents (*n* = 4), incidents during preparing or administering medicines (*n* = 3), and exposure to chemicals during the treatment of patients (*n* = 1). In laboratory settings (9 within the health sector, 1 in a school, and 20 in other types of laboratories), exposure is frequently the result of the release of gases and inadequate ventilation (*n* = 15) or inadvertent spillage of chemicals (*n* = 8). Incidents that have the potential to be serious due to the release of toxic gases (ammonia, hydrogen sulfide, or nitric oxides) in the agricultural sector are those involving slurry pits (*n* = 4). A large number of incidents in the business class culture, sports and recreation, occurred in swimming pools or wellness centers (*n* = 14), with guests and employees being exposed to chlorine gas as a result of errors in the mixing of cleaning agents. 

It is not uncommon for public sector workers (such as police officers, ambulance workers, firefighters, and military personnel) to be exposed to hazardous substances in the course of their duties responding to industrial or domestic incidents; for example, in some cases, police officers were exposed to chemicals during a raid on an illegal drug laboratory (*n* = 5).

### 3.7. Health Effects

Table A1 presents the total number and percentage of symptoms as reported to the DPIC at the time of consultation. It should be noted that there is limited information on the course of the health effects over time, as the Dutch Poisons Information Center does not routinely perform follow-up of all cases.

In one-quarter of patients, information on symptoms was lacking. At the time of consultation, 15% of patients (*n* = 281) were asymptomatic. Patients mainly developed effects involving the respiratory tract, such as dyspnea (12%), irritation of mucous membranes (11%), and cough (7%); the central nervous system (headache (20%) and dizziness (10%)), and gastrointestinal tract (e.g., nausea (12%) and pain in mouth or throat (7%)). 

The majority of patients reported mild to moderate symptoms. Only a small number of patients exhibiting potentially severe symptoms were reported to the DPIC; these symptoms included syncope (*n* = 12), coma (*n* = 8), and stridor (*n* = 4). Two fatalities were reported, one resulting from inhalation-related exposure to methanol in an illicit drug laboratory [33] and the other from inhalation-related exposure to an unidentified substance in a container of wood pellets.

## 4. Discussion

The growing use of chemicals in modern society has resulted in an increased risk of human exposure to and harm from hazardous substances [1,7,18,22,24]. While HAZMAT incidents may initially appear to be relatively minor, they frequently affect a considerable number of individuals, including workers, first responders, and the general public. 

In the Netherlands (which has approximately 18 million inhabitants), 516 HAZMAT incidents were reported to the DPIC between 2016 and 2023. Victims were most frequently exposed to gases or vapors (such as smoke, carbon monoxide, hydrogen sulfide, chlorine gas, methanol and ammonia) by inhalation. The potential for gases and vapors to spread rapidly in the environment increases the risk of exposure for a greater number of individuals. Other studies also describe substances such as carbon monoxide, ammonia and chlorine gas as common chemicals involved in acute toxic substance incidents [21,34,35]. In our study, carbon monoxide, ammonia, and chlorine gas were involved in 8.9%, 4.8%, and 4.8% of the incidents, respectively, which was in accordance with previous surveillance studies; Melnikova et al. described carbon monoxide (8.4%), ammonia (6.7%) and chlorine gas (2.5%) as chemicals commonly involved in fixed-facility incidents [35].

In our study, most incidents occurred in fixed facilities (88%), with lesser incidents occurring during transport (10%). A number of studies using data from the United States, Australia and Europe have reported similar results, with 64–72% of incidents occurring in fixed facilities, and 23–36% of these being transport-related [7,20,21,34,35]. As evidenced by the findings of our study and other similar studies, the release or spill of chemicals was more common than other exposure scenarios (82% in our study compared to 40–85% in the other studies) such as fires or explosions (13% in our study compared to 1–15% in other studies) [7,20,35]. 

Unsurprisingly, the majority of HAZMAT incidents reported to the DPIC occurred in the industrial (20%) and transport and storage sectors (12%), given the large number and variety of chemicals manufactured, transported, and used in these business classes. For the prevention of incidents in industry and the reduction of risks to human health, the focus should be on identifying the hazards [1,36]. Examples of hazards identified in this study include lack of information about the specific chemicals used, incorrect application of protocols, incorrect use of the correct PPE, and inadequate maintenance of equipment. Future incidents could be prevented by addressing these issues; for instance, through better inspection and enforcement.

In the transport and storage sector, we showed that a significant number of incidents (*n* = 11) were caused by the release of phosphine gas after fumigation of cargo. In order to ensure the safety of personnel, it is essential to implement preventative measures. In particular, these measures should include the enhancement of knowledge regarding the health hazards associated with pesticides, and the implementation of safety procedures when handling fumigated cargo [37].

It is remarkable that a significant number of incidents in our study occurred in other business classes, such as laboratories (6%); culture, sports, and recreation (3%); and education (3%). Although the handling of hazardous substances is a common practice in laboratory settings, and employees should be aware of the risks involved, incidents do occur with some regularity. Incidents in swimming pools and schools often involve a large number of victims. In the event of a chemical incident involving a large group of patients, symptoms could also be partly caused by psychological factors. Mass psychogenic illness (mass hysteria) is defined as the spread of illness signs and symptoms within a group without a clear physical or environmental cause. This phenomenon occurs as a result of a perceived threat that provokes a state of anxiety, which can, for example, be elicited by the presence of a noxious odor [38,39,40,41].

Public sector workers (such as police officers, ambulance workers, firefighters, and military personnel) could, in addition to other dangers, potentially be at risk of exposure to hazardous substances in the course of performing their duties in response to industrial or domestic incidents. This was observed in our study, where 9% of cases involved a public sector worker. It is vital that all first responders are adequately trained, including in the use of appropriate PPE, and possess a fundamental understanding of hazardous materials in order to recognize and avoid exposure [14]. Nevertheless, ambulance workers and hospital staff have a limited risk of serious exposure to hazardous substances during the treatment of chemically contaminated patients. De Groot et al. describe these minimal risks of secondary exposure for emergency workers and conclude that normal hygienic precautions (gloves and water-resistant gowns) will adequately protect hospital staff [42]. 

Poison Control Centers (PCCs) can play various roles in HAZMAT incidents. First, PCCs help in the triage of potentially intoxicated patients and provide guidance on their treatment. Comprehensive clinical toxicological knowledge of PCCs of hazardous chemicals can help improve assessments of health risks during incidents and prevent unnecessary hospital care. As in the Netherlands, in most countries’ PCCs provide direct medical advice to healthcare professionals, thereby playing an important role in the acute phase of HAZMAT incidents. Second, Poison Center data could provide a valuable source of information on HAZMAT incidents, complementing data from other organizations that collect and report information on these type of incidents (such as government agencies, public health, or occupational health/medical organizations). Together, this information can provide greater insight into the risks of HAZMAT incidents, such as identifying business classes, workers, and work processes at risk of HAZMAT incidents and the hazardous substances that are frequently involved. This information can help to improve preparedness for and prevention of chemical incidents; for instance, by raising awareness of hazardous substances and developing specific protocols for the different business classes. Based on the information from this study, examples of such protocols include providing clear working instructions to pool staff regarding the correct mixing of chemicals to prevent the release of chlorine gas; raising awareness among farmers about the risks of exposure to toxic agricultural gases (ammonia, hydrogen sulfide, or nitric oxides) and the use of PPE; and emphasizing the importance of following working instructions and ensuring proper ventilation in laboratories.

Our study has a number of limitations. Firstly, the data are derived from voluntary reports to the DPIC. As a result, the actual number of HAZMAT incidents is underestimated. Secondly, the DPIC is often consulted about a single patient, resulting in a lack of information about the actual scale of the incident and other patients involved. Thirdly, it is expected that the DPIC will be consulted primarily for cases requiring medical attention. This may result in an under-reporting of asymptomatic cases. Finally, our center does not typically perform follow-up. As a result, there is usually limited information about the overall health outcomes and treatments following DPIC consultation.

## 5. Conclusions

Surveillance data are an invaluable resource for public health policies in the context of incident preparedness and HAZMAT response. Poison Center data can assist in the identification of high-risk substances, business classes, and root causes of HAZMAT incidents. This information can be employed in the development of risk mitigation strategies to prevent future incidents.

## Figures and Tables

**Figure 1 ijerph-22-00158-f001:**
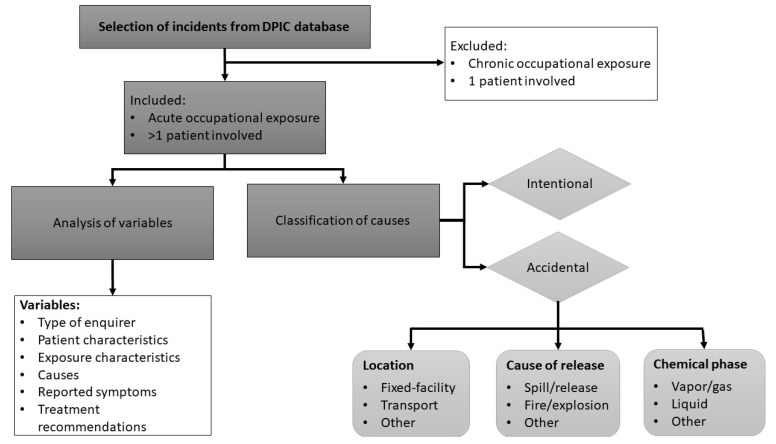
Flowchart of methodology used.

**Figure 2 ijerph-22-00158-f002:**
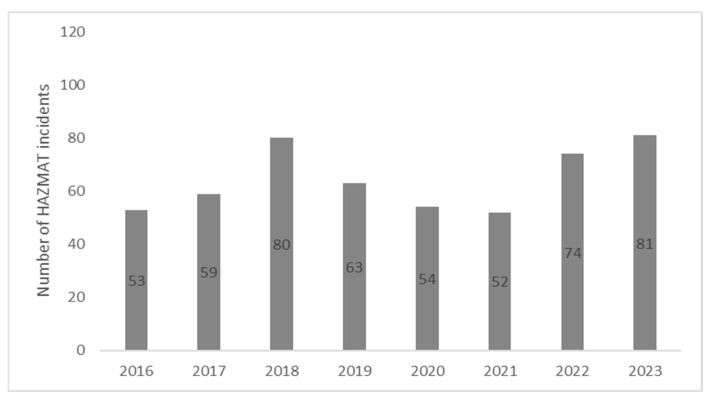
Number of acute occupational HAZMAT incidents reported to the Dutch Poisons Information Center (2016–2023).

**Figure 3 ijerph-22-00158-f003:**
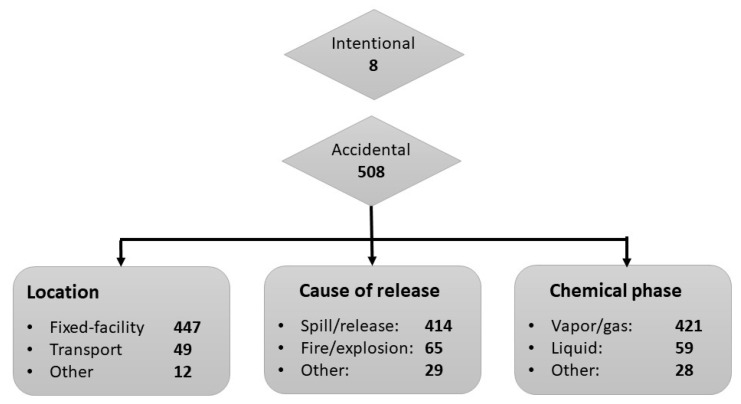
Characteristics of acute occupational HAZMAT incidents reported to the Dutch Poisons Information Center (2016–2023).

**Table 1 ijerph-22-00158-t001:** Most common substances involved (total number of exposures per substance group and the number of exposures to key representatives of the substance group) in HAZMAT incidents reported to the Dutch Poisons Information Center (2016–2023).

Substance Group	Substance	Exposures *n* = 1003 ^a^
Substance (group) unknown		111
Chemical asphyxiants		156
	Smoke ^b^	68
	Carbon monoxide	46
	Hydrogen sulfide (gas)	33
	Hydrogen cyanide (gas)	10
Acids		151
	Sulfuric acid	58
	(Per)acetic acid	14
	Hydrogen fluoride	10
	Nitric acid	9
Alcohols		46
	Methanol	26
	Isopropyl alcohol	6
	Ethanol	5
Petroleum products		40
	Naphthalene	16
	Hydraulic oil/motor oil	10
Chlorine compounds		38
	Chlorine gas	25
	Sodium hypochlorite	6
Alkalis		38
	Sodium hydroxide	21
	Sodium carbonate	4
Cyclic hydrocarbons		38
	Benzene	10
	Toluene	7
Pulmonary irritant gases		38
	Ammonia gas	25
	Sulfur dioxide	9
Metals and metal salts		34
	Mercury compounds	9
	Titanium compounds	4
Pesticides		31
	Phosphine	11
	Pyrethroids	5
Disinfectants		30
	Peroxides	17
	Quaternary ammonium compounds	13
Aldehydes and ketones		22
	Formaldehyde	11
Glycols and glycolethers		22
	(Di)propylene glycol	7
Phenols		16
	Phenol	12
Other		192

^a^ The number of exposures (1003) is higher than the number of incidents (516), as in some incidents, multiple substances were involved. ^b^ In 11 incidents, the smoke was the result of a lithium-ion battery fire.

**Table 2 ijerph-22-00158-t002:** Business classes in which the HAZMAT incidents reported to the Dutch Poisons Information Center (2016–2023) occurred.

Business Class	Total (*n*)	(%)
Industry	104	20.2
Transport and storage	62	12.0
Health and welfare care	49	9.5
Building and installation industry	48	9.3
Public services (e.g., police, ambulance, fire service, military)	45	8.7
Wholesale and retail	31	6.0
Laboratories (e.g. industry, health sector, schools)	30	5.8
Professional/corporate services (e.g., cleaning, security service, office)	26	5.0
Agriculture	22	4.3
Mining of minerals (e.g., oil, natural gas)	15	2.9
Culture, sports, and recreation (e.g., swimming pools, wellness centers)	15	2.9
Accommodation, provision of meals and drinks	14	2.7
Education	13	2.5
Other	12	2.3
Unknown	30	5.8

## Data Availability

The participants of the study did not give written consent for their personal data to be shared publicly, so supporting data are not available.

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
