# Peer review of "Poison Center Surveillance of Occupational Incidents with Hazardous Materials (2016–2023): Insights for Risk Mitigation and Incident Preparedness"

_ijerph, 2025, doi:10.3390/ijerph22020158_

Round 1

Reviewer 1 Report

Comments and Suggestions for Authors

This descriptive analysis of referred poison control cases from the Dutch Poisons Information Center (DPIC) is a useful addition to the literature. The rationale for this analysis and the data source are appropriately described. The results are thorough in presenting the findings from the DPIC case files.

I would ask the authors to consider revising two aspects paragraph four of the discussion. First, it is suggested that the first line of the paragraph be separated to form its own paragraph that focuses on findings from the industry, transportation, and storage sectors. In this expanded paragraph, the authors could draw attention to examples where better training and information for workers would be important to prevent poisonings. Given that the largest percent of cases were from these sectors, specific examples of opportunities for prevention are lacking in the discussion. This revision would address this latter point as well.

With information on industry, transportation, and storage sectors expanded and separated from the remaining material of the original fourth paragraph, this would address a second concern. This is about the information referring to a potential psychogenic component to some cases. This point is based on one reference (ref. 18: Jones et al. 2000) for psychogenic aspects that may be more relevant to adolescent population based behavior, with possible relevance to exposures in communal settings such as public swimming pools. This reference is not an ideal basis for suggesting relevance to occupational contexts, which is a conclusion that might be drawn from the way in which the current paragraph is structured. Revising these sections in the discussion as recommended would provide clarity on these issues.

I appreciate the opportunity to review this manuscript and commend the authors for compiling and presenting this informative set of data. 

Author Response

  1. Comment reviewer: First, it is suggested that the first line of the paragraph be separated to form its own paragraph that focuses on findings from the industry, transportation, and storage sectors. In this expanded paragraph, the authors could draw attention to examples where better training and information for workers would be important to prevent poisonings. Given that the largest percent of cases were from these sectors, specific examples of opportunities for prevention are lacking in the discussion. This revision would address this latter point as well.
    Response from the authors: Additional information on findings from the business classes "industry" and "transport and storage" has been added to the Results and is subsequently discussed in the Discussion. Our study also identified several hazards and opportunities for prevention, which have been included in the discussion of the revised manuscript.

  2. Comment reviewer: With information on industry, transportation, and storage sectors expanded and separated from the remaining material of the original fourth paragraph, this would address a second concern. This is about the information referring to a potential psychogenic component to some cases. This point is based on one reference (ref. 18: Jones et al. 2000) for psychogenic aspects that may be more relevant to adolescent population based behavior, with possible relevance to exposures in communal settings such as public swimming pools. This reference is not an ideal basis for suggesting relevance to occupational contexts, which is a conclusion that might be drawn from the way in which the current paragraph is structured. Revising these sections in the discussion as recommended would provide clarity on these issues.
    Response from the authors: The authors agree with the reviewer that reference 18 by Jones et al. is not an ideal basis for describing the role of mass hysteria in an occupational setting in general. Therefore, three other references with respect to occupational mass psychogenic illness have been added and the text was rephrased to: “In the event of a chemical incident involving a large group of patients, symptoms could also be partly caused by psychological factors. Mass psychogenic illness (mass hysteria) is defined as the spread of illness signs and symptoms within a group without a clear physical or environmental cause. This phenomenon occurs as a result of a perceived threat that provokes a state of anxiety, which can for example be elicited by the presence of a noxious odour.”

Reviewer 2 Report

Comments and Suggestions for Authors

1. Lines 10-24 The methodology should be explained to make it easy to understand to readers what kind of methodology is used to determine the used data.

2. Lines 45-49 Literature to support the statement is missing.

3. The introduction does not present a clear gap between the field and possible research fields or assumptions.

4. Lines 107-112 What is the meaning for the study of these variables?

5. Lines 116-120 What is the importance of this data in the research?

6. Figure 2, What is the meaning of each level? why the exposure was classified into levels?.

7. The discussion section should present in detail each numerical result to support each statement and then compare it with existing studies

Author Response

  1. Comment reviewer: Lines 10-24. The methodology should be explained to make it easy to understand to readers what kind of methodology is used to determine the used data.
    Response from the authors: The authors provided a more thorough explanation of the data collection method in the abstract. Consequently, this resulted in the word limit of 200 being exceeded by a few words (n=222).

  2. Comment reviewer: Lines 45-49. Literature to support the statement is missing.
    Response from the authors: As all data refer to reference 1, the authors moved this reference to the correct position in the introduction (line 59).

  3. Comment reviewer: The introduction does not present a clear gap between the field and possible research fields or assumptions.
    Response from the authors: Further details were included to outline the existing research gap and how poison center data can complement standard surveillance data (lines 70-75). Firstly, to emphasise the importance of surveillance, we have included literature that supports this. Secondly, the following text has been added: “The DPIC receives a significant number of calls on occupational incidents with relatively minor health consequences that are not included in regular Dutch injury statistics and are therefore considered supplementary [16]. Although these incidents may appear minor, they can be precursors to more significant incidents and provide an opportunity to learn from. Furthermore, these data can also be valuable for surveillance purposes.”

  4. Comment reviewer: What is the meaning for the study of these variables?
    Response from the authors: These variables were studied to summarise the characteristics of our dataset. The text was changed to clarify this (lines 132-136).

  5. Comment reviewer: Lines 116-120 What is the importance of this data in the research?
    Response from the authors: The information on the total number of calls to the DPIC for all categories of human exposure is, on reflection, not very relevant for this study and is therefore deleted (in text and figure).

  6. Comment reviewer: Figure 2, What is the meaning of each level? why the exposure was classified into levels?
    Response from the authors:
    In order to provide a more detailed description of the scenarios and to identify potential risk factors, the incidents were categorised. In the method section, the authors included that the classification system was in part based on the Hazardous Substances Emergency Events Surveillance (HSEES) of the US Agency for Toxic Substances and Disease Registry (ATSDR).
    Furthermore, the terms “level” and "four-level system" have been removed to avoid confusion. The use of "level" suggests a tiered approach, which has not been applied.
    Besides the adjustments in the methods section, Figure 3 of the revised manuscript has also been modified to better reflect the method used in this study.

  7. Comment reviewer: The discussion section should present in detail each numerical result to support each statement and then compare it with existing studies.
    Response from the authors: The authors incorporated numerical findings from their own research, as well as relevant data from other studies, in order to compare the results. See the highlighted text in the Discussion.

Reviewer 3 Report

Comments and Suggestions for Authors

This manuscript is about the Poison center surveillance of occupational incidents with hazardous materials (2016-2023): insights for risk mitigation and incident preparedness.

This manuscript needs a lot of improvement... a low-quality research.

Introduction

-Line 28-33: This section needs references....add

Incidents involving the release of hazardous materials (HAZMAT incidents), such as chemical spills, leaks, fires and the intentional release of toxic substances, have the potential to cause significant adverse effects on human health and the environment. HAZMAT incidents often have an occupational origin. In the event of such incidents, not only workers but also first responders and the general public could be at risk of harm.

-Improve literature review by adding 40 additional references related to topic to improve extensive literature review.

-Can you please establish the research gap in simple lines [Please add]

Materials and Methods:

- Research framework should be there to elaborate pattern of research. Add framework-flowchart and write this section in stepwise pattern.

Results:

-In this section add comparative analysis of your results with previously published papers (4-5 reference results)

-Please compare the results of this research within existing published research with existing materials. Line or bar charts can be added. [Please add]

Discussion:

Add something for field professionals. [Please add]

Limitations of the study

Please add as heading about the limitations of the study.

Author Response

  1. Comment reviewer: Introduction. Line 28-33: This section needs references....add.
    Response from the authors: In order to emphasise the potential severity of chemical incidents with respect to human health, the environment and society, the first paragraph of the introduction has been extensively revised (see highlighted text) and a number of relevant references have been included.

  2. Comment reviewer: Improve literature review by adding 40 additional references related to topic to improve extensive literature review.
    Response from the authors: We expanded our literature review and found an additional 15 articles that were relevant for this topic. These have been added to the introduction. In addition, another 8 references were added to the method and discussion of our revised manuscript..
    In the “Introduction”, additional information and references are included about the impact of HAZMAT incidents on human health, the environment and society, and the importance of surveillance. In the "Method" section, the authors included extra information and references about the system used to classify incidents. In the “Discussion” additional information and references were added on “Mass psychogenic illness” and incidents with fumigated cargo.

  3. Comment reviewer: Can you please establish the research gap in simple lines
    Response from the authors: Further details were included to outline the existing research gap and how poison center data can complement standard surveillance data (line 70-75). Firstly, to emphasise the importance of surveillance, we have included literature that supports this. Secondly, the following text has been added: “The DPIC receives a significant number of calls on occupational incidents with relatively minor health consequences that are not included in regular Dutch injury statistics and are therefore considered supplementary [16]. Although these incidents may appear minor, they can be precursors to more significant incidents and provide an opportunity to learn from. Furthermore, these data can also be valuable for surveillance purposes.”

  4. Comment reviewer: Materials and Methods: Research framework should be there to elaborate pattern of research. Add framework-flowchart and write this section in stepwise pattern.
    Response from the authors: A flowchart describing the method has been added to visualize the pattern of research.

  5. Comment reviewer: Results: In this section add comparative analysis of your results with previously published papers (4-5 reference results). Please compare the results of this research within existing published research with existing materials.
    Response from the authors: In the “Discussion” the authors incorporated numerical findings from their own research, as well as relevant data from other studies, in order to compare the results. See the highlighted text in the Discussion.

  6. Comment reviewer: Discussion: Add something for field professionals.
    Response from the authors: Information on the hazards and opportunities for prevention identified in our study have been included in the discussion of the revised manuscript (lines 292-297).

Round 2

Reviewer 2 Report

Comments and Suggestions for Authors

The introduction should include more numerical data from the literature  to present a clear gap between the field and possible research fields or assumptions

Author Response

The authors have included numerical data from the literature [24, 25]. The text has also been revised to more accurately describe the research gap.The revised text for the two paragraphs in the introduction on this subject is as follows:
“Surveillance data offers crucial insights into the prevalence and circumstances of HAZMAT incidents, which are essential for effective health and safety management [17-22]. In the Netherlands, multiple data sources document work-related fatalities and injuries [23-27]. For instance, VeiligheidNL reported that in 2017, an estimated 48,200 pa-tients visited an emergency department following a workplace injury. Among these inci-dents, only a minority involved chemical exposure (e.g., skin, eye; approximately 600 ED visits) or intoxication (approximately 500 ED visits). However, the causes of these expo-sures were not detailed [25]. Additionally, the Dutch Institute for Public Health and the Environment (RIVM) analyzes the nature, scale, and causes of incidents at companies handling large quantities of dangerous substances (Seveso facilities). Between 2019 and 2021, fourteen incidents involved the release of hazardous substances. One person likely sustained permanent injuries from a chemical burn, while other victims (21 in 6 incidents) suffered temporary injuries such as breathing problems, skin irritation, and burns [24].
Despite the presence of several authorities in the Netherlands that report on occupa-tional incidents, a comprehensive surveillance system that reports information on all types of occupational incidents involving hazardous materials (HAZMAT incidents) is currently lacking. The DPIC receives numerous calls about occupational incidents, in-cluding those with relatively minor health consequences that are not included in regular Dutch injury statistics and are therefore considered supplementary [16]. Although these incidents may seem minor, they can be precursors to more significant incidents and pro-vide valuable learning opportunities. Furthermore, these data can also be beneficial for surveillance purposes.”

Reviewer 3 Report

Comments and Suggestions for Authors

The authors have worked on my given comments. I suggest acceptance in this shape.

Author Response

The authors would like to thank the reviewer for the recommendation that the article can be accepted in this form.